# Domestication of Chili Pepper Has Altered Fruit Traits Affecting the Oviposition and Feeding Behavior of the Pepper Weevil

**DOI:** 10.3390/insects12070630

**Published:** 2021-07-12

**Authors:** Yosra Chabaane, Muhammad Haseeb, Betty Benrey

**Affiliations:** 1Laboratory of Evolutionary Entomology, University of Neuchâtel, 2000 Neuchâtel, Switzerland; yosra.chabaane@unine.ch; 2Center for Biological Control, College of Agriculture and Food Sciences, Florida A&M University, Tallahassee, FL 32307-4100, USA; muhammad.haseeb@famu.edu

**Keywords:** *Anthonomus eugenii*, oviposition, feeding behavior, chili domestication, plant traits, wild chilies

## Abstract

**Simple Summary:**

The pepper weevil is an economically important pest that causes major damage to fruits of chili pepper varieties selected for consumption. However, the impact of this pest on wild and ornamental peppers remains unknown. Therefore, we studied the effect of chili domestication on the feeding and oviposition behavior of pepper weevil when exposed to wild chili, ornamental varieties, and varieties used for consumption. More specifically, we examined how changes in fruit and flower size, fruit thickness, spiciness level, and fruit position as a result of the domestication of chili peppers affected their susceptibility to this specialist pepper pest. In addition, we recorded that fruits and flowers from wild and ornamental plants were less susceptible to pepper weevil attacks than those from chili varieties selected for consumption. Our results have important implications for chili pepper breeders and could guide the selection of new resistant varieties against this pest.

**Abstract:**

The pepper weevil, *Anthonomus eugenii*, Cano (Coleoptera: Curculionidae), is one of the most destructive pests of chili pepper. It causes extensive damage on varieties selected for consumption. However, the occurrence of this pest on wild and ornamental peppers remains unknown. We investigated the consequences of chili domestication on the feeding and oviposition of *A. eugenii* on fruits and flowers. We used plants of one wild accession, Bird Eye Pepper, five ornamental varieties (Pops Yellow, Black Pearl, Sedona Sun, Chilli Chilli, and Salsa Deep), and two domesticated varieties selected for consumption (Scotch Bonnet and Jalapeño). First, we characterized the plants according to their fruit and flower sizes, pericarp thickness, capsaicin level, fruit position, and flower color. Then, we evaluated the susceptibility of fruits and flowers to *A. eugenii*. Overall, domestication increased fruit and flower sizes and pericarp thickness, altered capsaicin levels, and altered fruit position and flower color. Weevils laid more eggs and caused more feeding damage on varieties selected for consumption than on wild and ornamental plants. Our results add to the growing literature on the consequences of crop domestication on herbivores. This knowledge could be integrated into breeding programs to select varieties resistant against the pepper weevil.

## 1. Introduction

The pepper weevil (*Anthonomus eugenii*, Cano) is a pepper (*Capsicum* spp.) specialist, although it is reported to attack other crops in the Solanaceae family such as eggplants (*Solanum melongena* L.) and the common black nightshade (*Solanum americanum* Mill.) [1,2]. It is a major pest of peppers in Mexico, its place of origin, as well as in the Caribbean and Southern United States, where it was first reported in Texas in 1904 [3,4,5]. Apart from its natural range, international trade has contributed to the spread of this pest in other countries (e.g., Canada, Netherlands, and Italy) [6,7,8]. Because of the widespread use of chili peppers in many countries [9], there is increasing concern that this pest could be inadvertently introduced worldwide. Females lay eggs singly in a cavity made with their rostrum (mouthpart), then seal the puncture with a brown fluid, secreted through the ovipositor, that hardens into a solid egg-plug (Appendix A) [5,10]. Upon hatching, the larvae feed on the seeds and soft tissue inside the developing fruit, whereas the adults feed on the flowers and young fruits (Figure 1) [6]. Females lay the maximum (3.1 eggs/day) on flower buds and/or on small fruits [11]. Their life cycle consists of an egg stage, three larval stages, pupal stage completed entirely inside the fruits, followed by the adult stage when *A. eugenii* exits the fruit by making a hole with their rostrum [1,5,12]. The typical damage of this insect is small holes on immature fruits, causing fruit deformation and premature fruit ripening and dropping [13].

The use of insecticides to control *A. eugenii* could be ineffective once the population is established in the field, because the entire development occurs inside the fruit [10,14]. Therefore, pest scouting methods using yellow sticky traps with a pheromone are recommended to establish the timing of pesticide application and reduce the adult population [15,16]. The use of natural enemies could offer an environmentally safe control method for this pest. In a survey conducted in Mexico [17], thirteen different species of parasitoids were reported to attack the pepper weevil. The main species found was *Catolaccus hunteri* Crawford (Hymenoptera: Pteromalidae), an ectoparasitoid of third larval stages. Parasitoid larvae develop externally on the larval host, and after two weeks, the adult parasitoids emerge [18]. However, the establishment of these parasitoids in new areas where the pepper weevil is introduced remains a challenge [19]. Therefore, additional pest management strategies are needed to control this pest in the open fields. Host plant resistance is an effective, economical, and ecofriendly method for pest control [20,21]. In this context, Berdegue et al. [22] explored the plant resistance against the pepper weevil of 23 virus-resistant lines that belong to cultivars of various *C. anuum* varieties, including Jalapeño, bell, pimiento, serrano, yellow, cayenne, long chile, tabasco, and cherry. They found that synchronous fruit production reduced the period of susceptibility to attack by *A. eugenii*. In another study, Seal and Martin [3] found that the highly pungent Habanero variety (*C. chinense* Jacquin) was more resistant to pepper weevil than nonpungent or mild *C. annum* cultivars, including Bell, Hungarian wax, or Jalapeño peppers. This suggests that the pungency level in fruits could be involved in plant resistance against this pest. However, this idea remains to be tested. 

The above studies used only chili varieties selected for fruit consumption. However, unlike many other crops, wild chili is also consumed either fresh or dry as condiment [23]. In Mexico and Southern US, wild chili plants grow in backyard gardens together with other domesticated plants (Chabaane, personal observation). In their natural habitat, wild chili provides an essential ecosystem service for local habitants who sell the dry fruits [24]. In addition to being domesticated for consumption, chili pepper has been selected for ornamental use. Ornamental peppers are morphologically diverse in fruit size, ripe fruit colors, foliar pigmentation that varies from green to purple, fruit and leaf shape [25]. In addition to their fruits, ornamental peppers were selected for different flower colors and size [26,27]. They were also selected for rapid seed propagation, heat, and drought tolerance [25,28,29]. When introduced to Europe in the 15th century, chili peppers were admired more as an ornamental plant than as a food source [25]. Today, they are gaining in popularity worldwide due to their beauty [29,30,31,32]. They were also reported as a potential alternative food source, known as a banker plant, for natural enemies of different pests such as the generalist mite *Amblyseius swirskii* or the insidious flower bug *Orius insidiosus* [33,34,35,36]. Despite the importance of wild and ornamental peppers, studies on the pepper weevil as a pest of chili peppers have essentially focused on varieties used for cooking. To date, there are no comparable studies that evaluate the impact of this pest on ornamental and wild chilies.

The suite of physiological and morphological traits that distinguish crops from their wild relatives is known as domestication syndrome [37]. For chili pepper, domestication has increased seed germination rate and fruit size, increased variation in fruit color, and changed fruit position from upright to pendant hidden by the foliage to reduce fruit predation by birds [38,39]. In addition, domestication influenced floral phenology, flower size and color, especially for varieties selected as ornamentals to make them more attractive [26,27]. The most characteristic trait of pepper is the capsaicinoids in fruits that are responsible for the spiciness or pungent taste [40]. Two capsaicinoids, capsaicin and dihydrocapsaicin, represent 90% of the whole capsaicinoids in fruits and are produced in the placenta [41]. In contrast to other crops for which domestication decreased secondary metabolites [42], species and varieties in the genus *Capsicum* have been selected for both increased and decreased levels of capsaicinoids [43,44] as compared to those of the wild ancestor, Chiltepin (*C. annuum* var. *glabriusculum*) [45,46]. A scale from low to high pungency (from 0 to 1,500,000 Scoville heat units = SHU), known as the Scoville chart, was developed to characterize chili varieties according to their spiciness [47]. According to the Scoville chart, Chiltepin has an intermediate pungency level (100,000–250,000 SHU) between highly pungent varieties (e.g., Habanero) and mild varieties (e.g., Jalapeño). 

The antifungal, antibacterial, and medicinal uses of capsaicin are very well-studied [48,49,50,51,52]. However, their effects on insects are poorly known, although it is commonly used as a pesticide [53,54]. In a previous study, we found that capsaicin reduced larval development, pupation, and adult emergence of the generalist herbivore (*Spodoptera latifascia*, Walker) and reduced the parasitism rate of its ectoparasitoid (*Euplecterus platyhypenae*, Howard) [55]. However, the effect of capsaicin on the specialist herbivore (*A. eugenii*) remains unclear. Moreover, it is still not known how altered traits and the purpose of domestication (for consumption or as ornamental) have affected the interaction between chili fruits and the pepper weevil.

In this study, we investigated the consequences of chili domestication on the feeding and oviposition behavior of the pepper weevil. To do so, we used one wild accession called Bird Eye Pepper, five ornamental varieties (Pops Yellow, Black Pearl, Sedona Sun, Chilli Chilli, and Salsa Deep), and two domesticated varieties selected for fruit consumption, Scotch Bonnet and Jalapeño. Our specific objectives were: (1) to examine how domestication has altered fruit size and pericarp thickness, fruit position, pungency level, and flower size and color; (2) to determine how these altered traits affect the feeding and oviposition of the pepper weevil. The results from this study will allow us, first, to evaluate the impact of chili domestication on the performance of this specialist pest and secondly, to identify plant traits potentially responsible for resistance against this insect that could be integrated in further breeding programs. 

## 2. Materials and Methods

### 2.1. Plants

#### 2.1.1. Wild Plants

Wild fruits of the Bird Eye Pepper (*Capsicum annuum* L. var. *glabriusculum)* were provided by Dr. Erdwin (FAMU, Tallahassee, FL, USA). The peppers are very small and grow spontaneously in backyard gardens from where they are collected, dried, or pickled in vinegar and used in many dishes. The fruits have an upright position on the plant, as in Appendix A. Although the *C. annum* pepper is originally from Mexico, the wild Chiltepin can also be found in Southern United States (Texas, Arizona, and Florida), where it is known as Bird Eye Pepper [56]. 

#### 2.1.2. Ornamental Plants

Plants were obtained from Tallahassee nursery https://www.tallahasseenurseries.com/ (accessed on 10 July 2021). We used the fruits of five ornamental varieties: Pops Yellow, Black Pearl, Sedona Sun, Chilli Chilli, and Salsa Deep. They differ in color, size, and form. They all have an upright position like the wild Bird Eye Pepper (Appendix A). They all belong to the species *Capsicum annum*. 

#### 2.1.3. Domesticated Plants for Fruit Consumption

We used three domesticated varieties that were selected for consumption and have been integrated into many cuisines, Jalapeño (*C. annum*), Pepperoncino (*C. annum*), and Scotch Bonnet (*C. chinense)*. They were collected at the Florida A&M University (FAMU) Research and Extension Center in Quincy and by a farmer, Dr. Bravo Brown in Cairo, Georgia. Pepperoncino fruits were purchased from the market in Neuchâtel, Switzerland. Fruits from Jalapeño and Scotch Bonnet are pendant and hidden by foliage (Appendix A). However, Pepperoncino fruits have an upright position (Appendix A).

The experiment on fruit size using the variety Peperoncino was conducted in Neuchâtel, Switzerland. All other experiments were conducted in Florida, where the pepper weevil is a major pest on chili peppers in open fields and greenhouses and where these plants are extensively grown [57,58].

### 2.2. Insects 

Jalapeño fruits with signs of *A. eugenii* infestation were collected from the field at the University of Georgia, Tifton Campus. Insects were kept inside rearing cages (30 cm × 30 cm × 30 cm) in an incubator at 28 ± 2 °C, 70 ± 5 ℅ RH, and 14:10 L:D period. The cages were checked daily for the collection of adults. 

### 2.3. Fruit Size and Pericarp Thickness Measurements

Fruit length, width and height, and pericarp thickness measurements were taken with an electronic digital caliper (Vogel, Kevelaer, Germany) with a measuring range from 0 to 150 mm and a resolution of 0.01 mm. Fruit size was calculated as follows: Fruit size (cm^3^) = [Length (mm) × Width (mm) × Hight (mm)]/100. 

### 2.4. Capsaicin Level in Fruits

The capsaicin level in fruits was characterized according to the pungency level reported in the literature and was based on the Scoville chart (Scoville Heat Unit = SHU) [47,59,60,61,62] (Figure 2). 

### 2.5. Effect of Domestication on Fruit Infestation by the Pepper Weevil

The aim of this experiment was to investigate whether the domestication status of chili fruits had an effect on the feeding and oviposition behavior of the pepper weevil. We conducted a no-choice test in Petri dishes (100 mm × 20 mm) with one wild accession (Bird Eye), five ornamental peppers (Pops Yellow, Black Pearl, Sedona Sun, Chilli Chilli, and Salsa Deep), and two varieties used for consumption (Jalapeño and Scotch Bonnet). We placed two fruits from each variety/accession with one couple of pepper weevils for one day. We had 10 replicates (Petri dishes) per treatment. After one day of infestation, we recorded the feeding holes and oviposition marks per fruit (a total of 20 fruits per treatment). 

### 2.6. Effect of Fruit Size on the Infestation by the Pepper Weevil 

To examine the effect of fruit size on feeding and oviposition, we conducted choice and no-choice tests with three categories of fruit size (small, medium, and large) with the same variety, Pepperoncino. For the choice test, we placed one fruit of each size category in the same Petri dish and for the no-choice test, we placed three fruits of the same size category in one Petri dish. We added one couple of weevils per Petri dish and counted the number of feeding holes and oviposition marks after one day of infestation. We had 10 replicates for each test (10 Petri dishes for the choice test and 30 Petri dishes for the no-choice test).

### 2.7. Effect of Domestication on Flower Infestation by Female Pepper Weevils 

As female weevils feed and lay eggs not only on the fruits but also on the flowers, we quantified the infestation on wild and domesticated flowers. We used flowers from the wild accession (Bird Eye), the ornamental pepper (Black Pearl), and the Jalapeño variety. Pepper weevils are also known to infest buds and flowers of other Solanaceae, so we included, as a comparison, the flowers of eggplant (*Solanum melongena* L.) [63]. Flower size was estimated using photos and a scale of 10 mm to evaluate the differences among treatments. Flowers of the wild pepper and Jalapeño are white, and the ornamental Black Pearl and eggplant flowers are purple. To study the feeding and oviposition of female weevils on the different flowers, one female was placed in a Petri dish with one flower (100 mm × 20 mm). We had five replicates (Petri dishes) per treatment. Flower infestation was evaluated at two time periods. The first period was 20 min after the release of the insect, and we referred to this as early infestation. During this period, we recorded the searching time, which is the time allocated by the mated female to find and start exploring the flower as well as the percentage of infested flowers (i.e., flowers with feeding marks) [(infested flowers/total number of flowers) ×100]. The second period was after 24 h (late infestation), we recorded again the percentage of infested flowers and the number of eggs laid per flower. 

### 2.8. Statistical Analysis

All statistical analyses were performed with R statistical software (version 3.5.3; R Development Core Team, 2020), using ANOVA, followed by residual analysis to verify the suitability of distributions of the tested models. Generalized Linear Mixed Models (GLMMs) with a Gaussian distribution, followed by *post hoc* analysis (Tukey’s), were used to compare the data on fruit and flower sizes, pericarp thickness, feeding holes in fruits, number of eggs in fruits and flowers, searching time on flowers, and number of infested flowers. Fruits and Petri dishes were included as random factors. The overdispersion of the data was verified and when necessary, the correction quasibinomial was applied. The sample size and number of replicates for all experiments are indicated directly in the figure captions.

## 3. Results

### 3.1. Effect of Chili Domestication on the Fruit Size, Pericarp Thickness, and Capsaicin Level

#### 3.1.1. Fruit Size

Overall, chili domestication increased fruit size (Figure 3a). Except for the three ornamental peppers, Pops Yellow, Black Pearl, and Sedona Sun, fruits of the other domesticated varieties were significantly larger than those of the wild Bird Eye Pepper (F_[7,247]_ = 23.609, d.f = 7, *p* < 0.001). Fruits domesticated for consumption were at least 35% larger than ornamental fruits, and the largest fruit size was recorded for the Jalapeño variety.

#### 3.1.2. Pericarp Fruit Thickness

Chili domestication increased pericarp thickness (Figure 3b). All ornamental and consumption varieties had a thicker pericarp compared to that of the wild Bird Eye Pepper (F_[7,247]_ = 30.279, d.f = 7, *p* < 0.001). The ornamental Salsa Deep Pepper and Jalapeño variety had the thickest pericarp, over 2.5 mm, which was three times more than the thickness recorded on wild fruits.

#### 3.1.3. Capsaicin Level in Fruits

Based on the pungency level reported in the literature and the Scoville chart (Figure 2), the characterization of chili pepper revealed that the pungency level for wild pepper (100,000–250,000 SHU) was intermediate, between those of the spicy Scotch Bonnet (200,000–350,000 SHU) and the mild Jalapeño (3500–8000 SHU), both varieties used for consumption. However, the capsaicin level in ornamental varieties was lower than in the wild pepper, within a gradient from 0 SHU (nonpungent Chilli Chilli variety) to around 30,000 SHU (Black Pearl variety).

### 3.2. Effect of Domestication on Fruit Infestation by the Pepper Weevil

#### 3.2.1. Feeding on Fruits

Overall, both the wild and ornamental varieties were less attacked by the pepper weevil as compared to the varieties selected for consumption (Jalapeño and Scotch Bonnet) (Figure 4a; F_[7,152]_ = 15.809, d.f = 7, *p* < 0.001). The only exception was Sedona Sun (ornamental) that had as many feeding marks (around 10 marks per fruit) when compared to Scotch Bonnet. Moreover, the pepper weevil fed 45% more on Jalapeño than on Scotch Bonnet. We did not find significant differences for the feeding marks between wild pepper and ornamental varieties.

#### 3.2.2. Oviposition on Fruits

Female weevils laid 34% more eggs on varieties used for consumption (Jalapeño and Scotch Bonnet) than on wild pepper or ornamental varieties (Figure 4b; F_[7,152]_ = 4.3715, d.f = 7, *p* < 0.001). However, Bird Eye Pepper and the ornamental variety Black Pearl had as many oviposition marks as Scotch Bonnet, but significantly fewer than Jalapeño. Varieties used for the same purpose of domestication (ornamental or consumption) had no significant differences for the number of eggs laid per fruit.

### 3.3. Effect of Fruit Size on the Infestation by the Pepper Weevil

#### 3.3.1. Choice Test

We found significant differences in fruit size for the three categories, small, medium, and large, shown in (Figure 5a; F_[2,27]_ = 151.94, d.f = 2, *p* < 0.001). We did not find significant effect of fruit size on feeding marks (Figure 5b; F_[2,27]_ = 0.3631, d.f = 2, *p* = 0.6989). However, females laid more eggs on small fruits than on large or medium fruits (Figure 5c; F_[2,27]_ = 13.968, d.f = 2, *p* < 0.001).

#### 3.3.2. No-Choice Test

The three categories used for this test had significantly different fruit size (Figure 5d; F_[2,87]_ = 273.28, d.f = 2, *p* < 0.001). We did not find significant effects of fruit size on feeding (Figure 5e; F_[2,87]_ = 0.0583, d.f = 2, *p* = 0.9434) or oviposition marks by the pepper weevils (Figure 5f; F_[2,87]_ = 1.448, d.f = 2, *p* = 0.2416).

### 3.4. Effect of Domestication on Flower Infestation by Female Pepper Weevils

#### 3.4.1. Flower Size

We found significant differences in flower size among the four treatments (F_[3,16]_ = 95.667, d.f = 3, *p* < 0.001). Flowers of the wild Bird Eye were the smallest (15.8 ± 0.9 mm), followed by the ornamental Black Pearl (43.6 ± 3.2 mm) and the variety selected for consumption, Jalapeño (73.4 ± 2.9 mm), and the largest flowers were from the eggplants (121.4 ± 8 mm).

#### 3.4.2. Searching Time

We did not find significant differences for the time spent by pepper weevils to find the flowers (Figure 6a; F_[3,16]_ = 43.891, d.f = 3, *p* = 0.7018).

#### 3.4.3. Infestation on Flowers

Early (after 20 min) and late infestation (after 24 h) on flowers are shown in Figure 6b. Within the first 20 min of the experiment, the infestation was 40% to 60% higher on the Jalapeño flowers than on wild and ornamental flowers. The eggplant flowers were all infested within 20 min (Figure 6b; χ^2^ = 16.738, d.f = 3, *p* = 0.01708 *). This effect was further confirmed after 24 h (Figure 6b; χ^2^ = 13.460, d.f = 3, *p* = 0.01187 *). The weevils attacked the wild (Bird Eye) varieties more than the ornamental flowers (Black Pearl) at the beginning of the experiment, but the infestation level was similar after one day.

#### 3.4.4. Oviposition on Flowers

We found significant differences for the number of eggs on flowers (Figure 6c; F_[3,16]_ = 1.22, d.f = 3, *p* = 0.01562 *). Female weevils laid eggs only on Jalapeño and eggplant flowers, and none on wild and ornamental flowers.

## 4. Discussion

Pepper weevil causes considerable damage to domesticated chili varieties used for consumption. The present study was therefore aimed at investigating the unknown effects of this pest on ornamental and wild peppers. Overall, we found that chili domestication has increased fruit size and pericarp thickness, altered capsaicin levels by selecting varieties with lower and higher capsaicin content than the wild chili, and changed the fruit position of varieties used for consumption. Additionally, we found that weevils preferentially fed and oviposited on varieties selected for consumption. Thus, our results revealed that domesticated traits in varieties used for consumption of their fruits have shown increased susceptibility to pepper weevil.

Several studies have shown that domestication has increased the susceptibility of crops to herbivore attack [64,65,66,67]. In the case of chili pepper, different physical traits were reported to play an important role in plant resistance against the pepper weevil. For example, Wu, Haseeb [57] showed that *A. eugenii* preferred to feed and oviposit on small-sized, thin-walled, and small-mass fruits over large-sized, thick-walled, and large-mass fruits from the same variety. In a choice test, we found similar results across varieties: females preferred to oviposit on small rather than medium-size or large fruit varieties, but not within the same variety; weevils fed equally on Pepperoncino fruits regardless of their size. Seal and Martin [3] also found that when offered different cultivars, weevils were more attracted to medium to large fruits (≥1.5 cm long) than to small fruits (<1.5 cm). Our results showed that Jalapeño and Scotch Bonnet, both consumption varieties, had the largest fruit size and were attacked by weevils the most compared to ornamental and wild peppers with smaller fruits. A plausible explanation is that beetles choose cultivars with large fruits that offer more resources for feeding and oviposition and possibly a lower risk of predation [68]. Conversely, Porter, Lewis [69] found that within the same variety (Jalapeño), weevils preferred small fruits. They suggest that small fruits resemble flower buds, where the first infestations take place [69]. Although we only had two varieties selected for consumption, chili fruits domesticated for this purpose are known to be larger than wild and ornamental fruits [25,70,71,72]. Indeed, Silva, Jasmim [70] point out that in the production of ornamental plants, breeders tend to favor small-sized fruits present in large quantities, traits that enhance their attractiveness and beauty.

Pericarp thickness has also been proposed as a plant trait involved in resistance against this pest [73]. In cotton, the boll wall thickness was responsible for reduced damage on cotton plants caused by the boll weevil (*Anthonomus grandis grandis*, Boheman), a sister species of the pepper weevil [74]. In our study, although domestication increased pericarp thickness, this did not seem to influence fruit susceptibility to pepper weevil, as varieties domesticated for consumption suffered the most damage. Moreover, most of the ornamental varieties had a pericarp as thick as that of Scotch Bonnet and Jalapeño fruits, but they were less damaged by the weevils. This suggests that other traits (e.g., fruit size) must be responsible for the increased susceptibility of consumption varieties. In addition, the maximum pericarp thickness recorded in our study was around 2.6 mm (Jalapeño and Salsa Deep varieties). This measurement is doubled and may reach over 6 mm for some bell pepper cultivars [75] and may thus explain their resistance to the pepper weevil [73]. The female weevil creates a cavity with her rostrum (with a mandibulated mouthpart) before depositing the egg and seals the puncture with a brown fluid that hardens and darkens into a solid egg-plug (Appendix A) [1,5,10]. The rostrum length for both sexes is around 1.5 mm (Appendix A). Therefore, a thick pericarp may hinder the feeding and oviposition of this insect. Future breeding programs could exploit this knowledge and select varieties with thicker pericarps. However, selecting varieties with a thicker pericarp may not always be compatible with the use of natural enemies for biological control. For example, the ability of the parasitoid *C. hunteri* to insert its ovipositor and parasitize the larvae of the pepper weevil decreases in fruits with a thicker pericarp [76]. Fruit position is another plant trait that was targeted by domestication [38,40]. The shift was from an upright position for wild chilies to pendant, hidden by foliage for many but not all domesticated plants [40]. For example, the fruits of the ornamental varieties used in our study kept the upright position similar to that of the wild Bird Eye (Appendix A). Therefore, we expect that these plants could be more exposed to insect and bird attacks compared to other varieties where fruits are pendant and hidden under leaves (e.g., Jalapeño and Scotch Bonnet). This hypothesis remains to be tested.

In contrast to other crops for which domestication aimed at reducing secondary metabolites [42], chili pepper was selected for both increased and decreased levels of capsaicinoids [43,44] as compared to that in the wild ancestor, Chiltepin [45,46]. Because in the current study we used only one wild chili accession, we should be cautious with our conclusions pertaining to the wild chili. However, in another study with nine Chiltepin populations collected in Southern Mexico, we found minimal variation in capsaicin levels among populations (Chabaane et al., unpublished data). Conversely, capsaicin levels vary greatly among the domesticated varieties (Figure 2) [47,55]. In insects, some studies with dried chili fruits and capsaicin-spiked artificial diet have demonstrated that capsaicin inhibits the feeding of *Henosepilachna vigintioctomaculata*, Motschulsky [77], deters oviposition of *Delia antiqua*, Meigen [78], and slows down larval development of *Earias insulana*, Boisduval [79]. The effects of capsaicin on generalist and specialist herbivores are expected to be different. For example, Ahn, Badenes-Pérez [80] reported that feeding on a capsaicin-spiked diet slowed the development of the generalist herbivores *Spodoptera frugiperda* JE Smith, *Heliothis virescens* Fabricius, and *Helicoverpa zea* Boddie, but did not affect the growth and survival of larvae of the tobacco budworm (*Helicoverpa assulta*), a specialist on Solanaceae. The latter can detoxify these secondary metabolites when feeding on hot pepper fruits [81,82]. In a previous study with fresh chili fruits, we also found that capsaicin had a negative effect not only on the generalist herbivore (*Spodoptera latifascia*, Walker) but also on its ectoparasitoid (*Euplecterus platyhypenae*, Howard), only at high doses [55].

Our results showed that the selection of fruits by the weevils based on capsaicin levels is unclear. Despite being a specialist, *A. eugenii* preferred to feed on and laid more eggs on Jalapeño fruits (mild) than on spicy peppers (e.g., Scotch Bonnet, Wild Bird Eye, and the ornamental Black Pearl). This preference for less pungent varieties was also reported by Seal and Martin [3]. They found that the highly pungent Habanero variety (*C. chinense* Jacquin) was less susceptible to *A. eugenii* than nonpungent or mild *C. annum* cultivars, including Bell and Jalapeño peppers [47]. However, in our study, both larvae and adults were also observed feeding on the fruit’s placenta of the Scotch Bonnet variety (Chabaane, personal observation), where capsaicinoids are concentrated [83]. Meanwhile, the ornamental hot pops yellow fruit, mild in pungency, and the nonpungent ornamental Chilli Chilli were more resistant to the weevil’s attack than the spicier chili varieties used for consumption (Figure 4a,b). As a specialist on chili pepper, it is expected that this pest is adapted to capsaicinoids, and both larvae and adults can tolerate the spiciness, but the mechanism remains unclear. We believe that it is more likely that other plant traits function as a defense against the pepper weevil.

For flowers, chili domestication increased the size and increased color variation, particularly for varieties selected as ornamentals, to make them more attractive for a commercial purpose [27]. We found that female weevils infested the larger flowers of Jalapeño and eggplant more. In contrast, flower color did not appear to have any effect on weevil infestation and oviposition. However, we should be cautious with this conclusion as we only used flowers from one representative of each category (wild, ornamental and for consumption, and eggplant as a comparison). In this study, we mainly focused on the consequences of chili domestication on morphological traits and capsaicin content, a direct chemical defense. Other traits like volatiles may also play a role in the host plant choice of the pepper weevil. Indeed, Addesso and McAuslane [63] showed that *A. eugenii* could detect and orient to constitutive host plant volatiles released from chili pepper fruits and flowers. Moreover, female weevils were attracted to damaged fruiting and flowering plants over undamaged plants, and preferred flowering and fruiting plants with actively feeding weevils over plants with old feeding damage [84]. Our results showed that both fruits and flowers of varieties used for consumption were more susceptible to these beetles than the ones from ornamental or wild plants. The extent to which fruit and flower volatiles play a role in this choice and whether domestication has altered these traits could be the subject of future research.

## 5. Conclusions

This study showed that chili domestication has altered morphological and chemical (capsaicin) traits in fruits with direct consequences for the feeding and oviposition by the pepper weevil. We focused on fruits and flowers mainly from the *C. annum* species (except for the Scotch Bonnet variety). Overall, we found that chili domestication increased fruit size and pericarp thickness, altered capsaicin levels, and changed the fruit position of varieties used for consumption. In addition, our results revealed that domesticated traits in varieties used for consumption have increased the susceptibility towards the pepper weevil. Accordingly, weevils preferred to feed on and laid more eggs on these varieties. Future work should examine other *Capsicum* species as well as additional plant traits (e.g., plant size, branching pattern, and volatile emissions) that might enhance resistance against this insect pest. For example, planting combinations of different chili varieties, including ornamental varieties and even wild chili plants, could create associational effects and reduce herbivory and disease transmission. A better understanding of the natural resistant traits of wild chili plants that have been altered as a result of domestication could help in the development of new varieties resistant to the pepper weevil and other pests.

## Figures and Tables

**Figure 1 insects-12-00630-f001:**
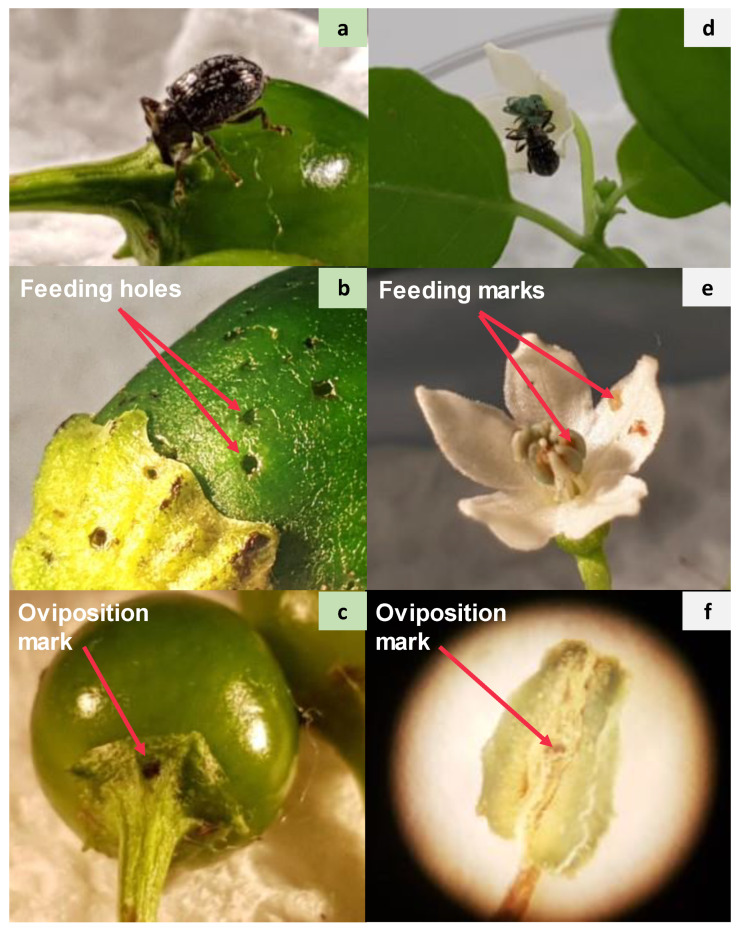
Pepper weevil damage on chili fruits (**a**–**c**) and flowers (**d**–**f**). All images © Y. Chabaane.

**Figure 2 insects-12-00630-f002:**
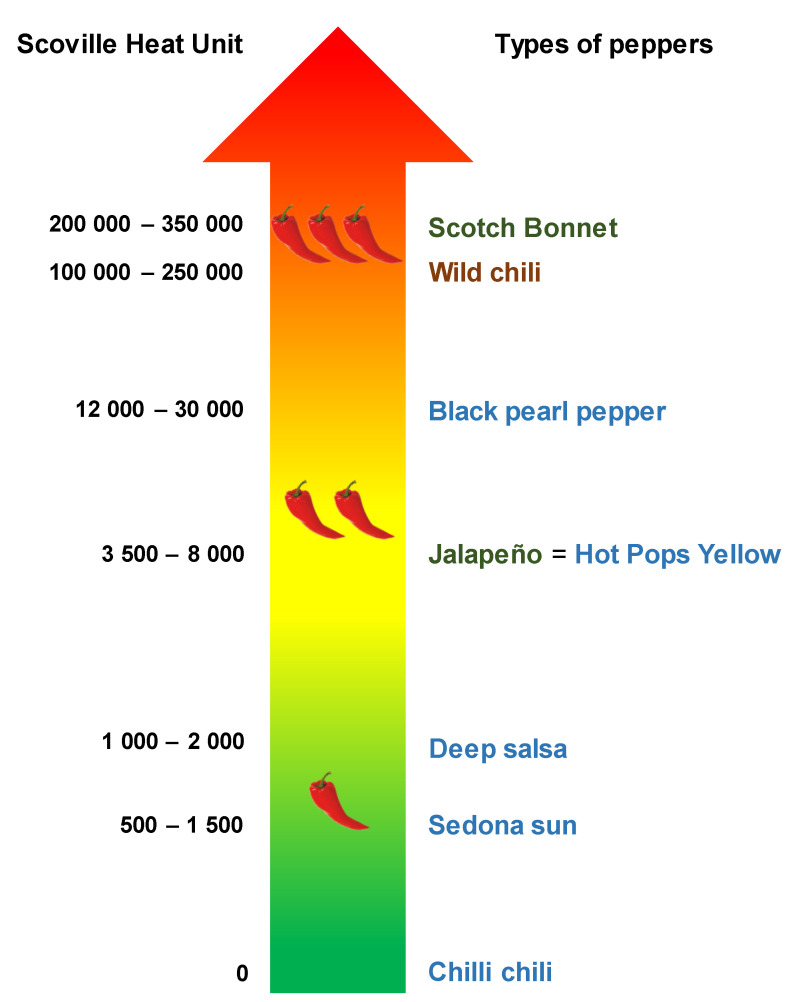
Characterization of capsaicin level in wild (Brown), domesticated as ornamental (Blue), and domesticated for consumption (Green) peppers according to the Scoville scale. © Y. Chabaane.

**Figure 3 insects-12-00630-f003:**
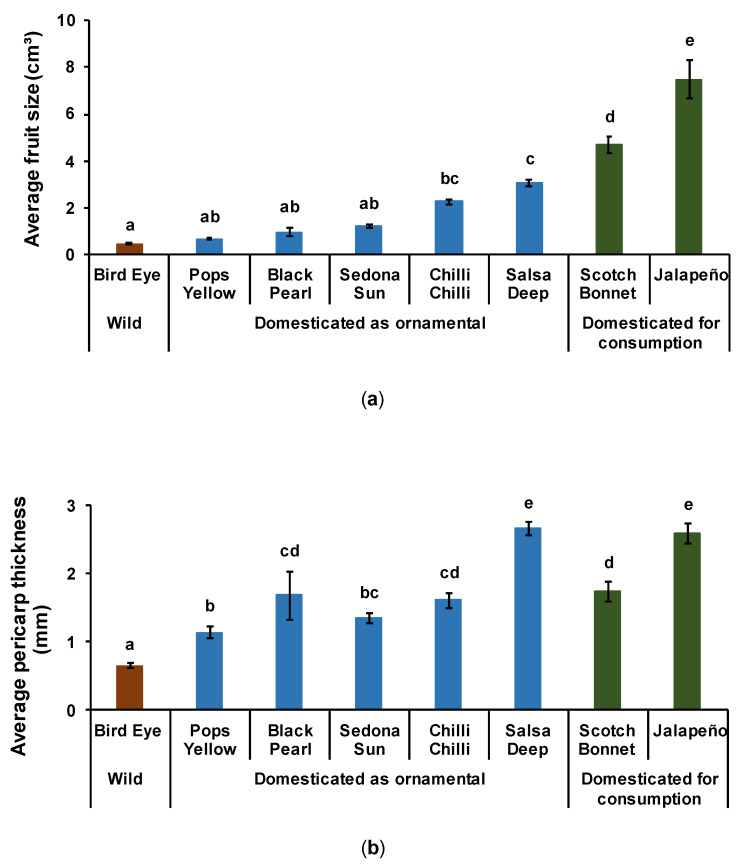
Effect of chili domestication on fruit size (**a**) and pericarp thickness (**b**). Sample size: Bird Eye (N = 25); Pops Yellow (N = 20); Black Pearl (N = 25); Sedona Sun (N = 30); Chilli Chilli (N = 20); Salsa Deep (N = 41); Scotch Bonnet (N = 32); Jalapeño (N = 62). The differences among treatments are indicated by different letters (F-test, Tukey’s post-hoc test with Bonferroni correction: *p* < 0.001).

**Figure 4 insects-12-00630-f004:**
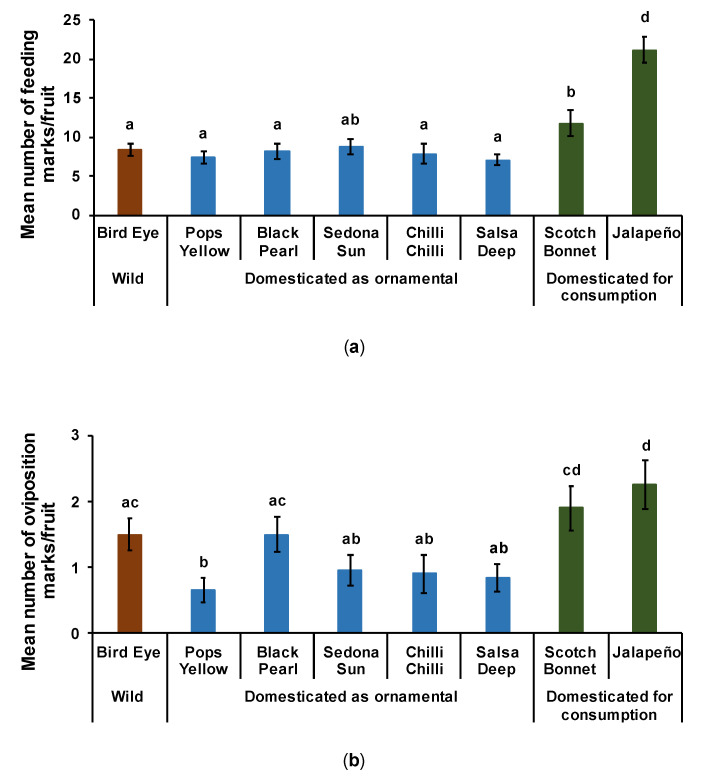
Infestation by pepper weevil on ornamental and domesticated peppers showing the mean number of feeding holes per fruit (**a**) and the oviposition marks per fruit (**b**). No-choice tests using two fruits per Petri dish. The differences among treatments are indicated by different letters (F-test, Tukey’s post-hoc test with Bonferroni correction: *p* < 0.001, N = 20).

**Figure 5 insects-12-00630-f005:**
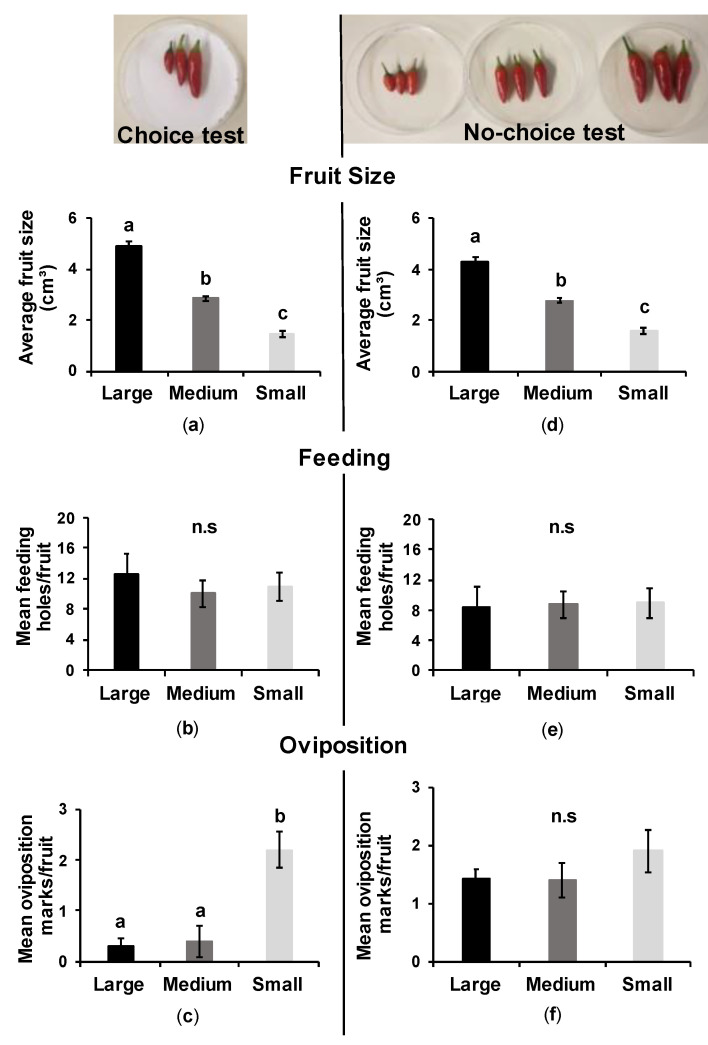
Effect of fruit size on the feeding and oviposition of the pepper weevil on Pepperoncino variety, using choice (**a**–**c**) and no-choice tests (**d**–**f**). The differences among treatments are indicated by different letters (F-test, Tukey’s post-hoc test with Bonferroni correction: *p* < 0.01, N = 10). All images © Y. Chabaane.

**Figure 6 insects-12-00630-f006:**
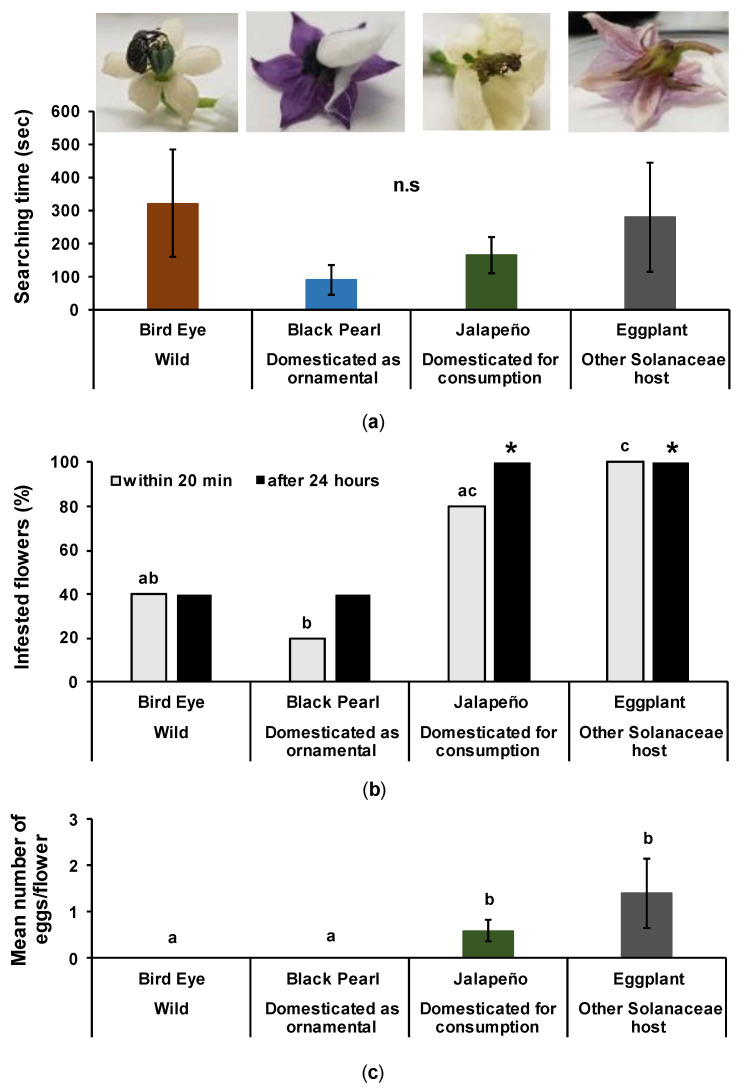
Pepper weevil attacks on flowers of wild pepper (Bird Eye), ornamental pepper (Black Pearl), pepper domesticated for consumption (Jalapeño), and eggplant. Searching time to find the flower (**a**), percentage of infested flowers (**b**), and mean number of eggs per flower (**c**). The differences among treatments are indicated by different letters for 20 min and stars (*) for 24 h (F-test (a, c) and Chi-test (b) followed by Tukey’s post-hoc test with Bonferroni correction: *p* < 0.05, N = 5). All images © Y. Chabaane.

## Data Availability

The data presented in this study are available on request from the corresponding author.

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
