# Peer review of "Domestication of Chili Pepper Has Altered Fruit Traits Affecting the Oviposition and Feeding Behavior of the Pepper Weevil"

_insects, 2021, doi:10.3390/insects12070630_

Round 1
Reviewer 1 Report
The manuscript is ready for publication. It has high quality.
The manuscript "Domestication of chili pepper has altered fruit traits affecting the oviposition and feeding behavior of the pepper weevil" has scientific merit and is presented as a complete manuscript of high scientific value. The characterization of the insect is robust and the authors did an excellent job in describing the injuries at the different stages of plant development. Perhaps the crop is not very popular when accessing the article, as it is restricted to production, but it is still an excellent article.
Author Response
We thank the reviewers for the positive feedback
Reviewer 2 Report
A well written manuscript with minor corrections, please read my comments and undertake required corrections. In page 5 line 179 Author/s mention Table 2.5 but there is no Table Also in Page 9 line 272 mentioned Table 0.N=20,however I did not find any tables in the manuscript you have to explain this.

Author Response
We thank the reviewer for the positive impression on our study, and we addressed the comments as follows:
In page 5 line 179 Author/s mention Table 2.5 but there is no Table -> Thank you for your comment.
The correct caption was added for figure 2 line 179 “Characterization of capsaicin level in wild (Brown), domesticated as ornamental (Blue), and domesticated for consumption (Green) peppers according to the Scoville scale. © Y. Chabaane”
Also in Page 9 line 272 mentioned Table 0.N=20,however I did not find any tables in the manuscript you have to explain this.
The correct caption was added for figure 4 line 272 -> ”Infestation by pepper weevil on ornamental and domesticated peppers showing the mean number of feeding per fruit (a) and the oviposition marks per fruit (b). No choice tests using two fruits per Petri dish. The difference among treatments is indicated by different letters (F-test, Tukey post-hoc test with Bonferroni correction: P<0.001, N= 20)”.

Reviewer 3 Report
Although the study is sound and well-presented, the conclusions are common sense and unsurprising. At the end of the day, there is no conclusion I can find to know what breeding changes to peppers would help to control pepper weevils.
Minor corrections/suggestions:
line 13 change precisely to specifically
line 30 change interest to literature
\line 44 change raising to increasing
line 183 a 100 mm X 20 mm petri dish seems too shallow to hold the peppers. Did they fit?
Figure 5 is a little confusing. Please revise to clarify the presentation
line 386 please provide insect species tested.
Author Response
All minor corrections are done in the revised manuscript.
line 183 a 100 mm X 20 mm petri dish seems too shallow to hold the peppers. Did they fit? YES, they did and with plenty of space to move around freely.
We have revised figure 5 to make it clearer attached the revised figure, and have provided the scientific names for insects (line 386)
We expanded in the conclusion specifying how we believe that our findings can be used in management practices against the pepper weevil and other pests.
